# The Role of Renewable Energies, Storage and Sector-Coupling Technologies in the German Energy Sector under Different CO₂ Emission Restrictions

**Arjuna Nebel \***[ID]**, Julián Cantor, Sherif Salim, Amro Salih and Dixit Patel**

Faculty of Process Engineering, Energy and Mechanical Systems, Cologne Institute for Renewable Energy, TH Köln, 50678 Köln, Germany
\* Correspondence: arjuna.nebel@th-koeln.de; Tel.: +49-221-8275-2239

**Abstract:** This study aimed to simulate the sector-coupled energy system of Germany in 2030 with the restriction on CO₂ emission levels and to observe how the system evolves with decreasing emissions. Moreover, the study presented an analysis of the interconnection between electricity, heat and hydrogen and how technologies providing flexibility will react when restricting CO₂ emissions levels. This investigation has not yet been carried out with the technologies under consideration in this study. It shows how the energy system behaves under different set boundaries of CO₂ emissions and how the costs and technologies change with different emission levels. The study results show that the installed capacities of renewable technologies constantly increase with higher limitations on emissions. However, their usage rates decreases with low CO₂ emission levels in response to higher curtailed energy. The sector-coupled technologies behave differently in this regard. Heat pumps show similar behaviour, while the electrolysers usage rate increases with more renewable energy penetration. The system flexibility is not primarily driven by the hydrogen sector, but in low CO₂ emission level scenarios, the flexibility shifts towards the heating sector and electrical batteries.

**Keywords:** sector coupling; renewable technologies; CO₂ cap; PyPSA; renewable energy penetration; multi objective optimisation

## 1. Introduction

The industrialised countries are the source of most past and current greenhouse gas emissions. Thus, they are expected to take the major share in cutting emissions on home ground [1,2]. The 2015 Paris Agreement recommends limiting global warming below 2, preferably to 1.5 °C, compared to pre-industrial levels, and that all developed countries set emission targets to reduce greenhouse gas (GHG) emissions [2,3]. The European Commission proposed, in September 2020, raising the 2030 GHG reduction target to at least (55%) compared to 1990 levels [2,4]. This goal is part of the long-term strategy of the European Union (EU) to become climate-neutral by 2050 and develop an economy with net-zero greenhouse gas emissions [4]. The key standing targets of the EU 2030 climate and energy framework, apart from reaching the increased target of (55%) emissions reduction, are also to achieve a 32% share for renewable energy and 32.5% improvement in energy efficiency [4].

In this regard, Germany set a roadmap in 2015 to reduce GHG emissions to 55%, compared to 1990 levels [5]. The power, heating and transport sectors are expected to decrease their CO₂ emissions. Figure 1 shows the configuration of primary energy consumption in 2015, their CO₂ emissions and one probable scenario for 2030. To accomplish the GHG emission reduction targets, adopting renewable energy sources requires a holistic transition across all sectors. In an energy system with high shares of intermittent renewable energies, periods are to be expected when the renewable generation exceeds the conventional load, resulting in demand being consistently surpassed and hence a surplus [6], which leads to

curtailment of generation. Curtailment refers to the reduction in the power output of the generators compared to what could be generated given the existing resources [7].

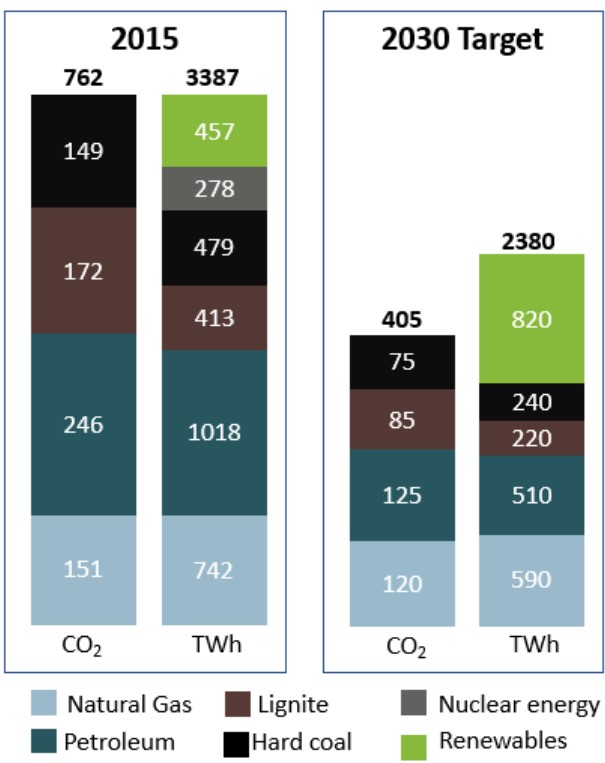

**Figure 1.** Energy-related greenhouse gas emissions in million tonnes of $CO_2$ and primary energy consumption in *TWh* in Germany for 2015 and a scenario taking the targets for 2030 into account. Data adapted from [5].

In 2020, 6146 GWh of electricity was curtailed in Germany because of feed-in management measures. This was a reduction of 5% compared to 2019 [8]. In 2019, the curtailment summed up to 6482 GWh and accounted for 2.9% of the renewable energy generated from installations qualified for payments under the German renewable energy sources act (EEG) [9]. The coupling of sectors is one strategy to reduce energy curtailments and make effective use of the energy surplus [10]. According to [6], sector coupling refers to the connection of the electricity sector with other sectors (buildings, transport and industry) by using different technologies, such as heat pumps, combined heat, power generators, electrical boilers or electric-driven vehicles.

The coupling of the electricity and the heat sectors has been studied in a huge variety of articles. Palzer and Henning created a model that met the electricity and heat demand with a share of up to 100% renewables for the entire building sector in Germany [11,12]. Abdur Rehman et al. studied the importance of heat pumps and thermal energy storage in reducing $CO_2$ emissions [13]. Bashir et al. presented a model optimising the coupling of the electricity and the heat sectors in Finland [14]. Robinius et al. analysed the national and international power and transport sector coupling [10], and presented a model that connects Germany's electricity and transport sector [15]. Traber et al. presented a scenario that shows an economically feasible 100% renewable energy system for 2030 in Germany by coupling the power sector with both the heat and the transport sectors [16].

Among the tools that are used to study sector coupling, Python for power system analysis (PyPSA) is an open-source Python tool, which has been developed and used to analyse complex power systems [17,18]. Its strength allows for the integration of renewable energies and other sectors such as heating and transport over multiple periods for operational and optimal investment [17].

PyPSA has been used to model energy systems at a national and multinational level to integrate renewables under $CO_2$ emission targets. In [19], the authors modelled the European coupled energy system, with defined $CO_2$ emissions targets and the penetration of solar and wind as leading technologies for the energy transition. The analysis concluded that about 33% of the demand in Europe by 2035, under a zero-$CO_2$-emissions constraint, will be covered by solar. It also shows the benefits of interconnecting European countries with transmission lines to use both wind and solar. It was found that the inclusion of electric vehicles and cooling technologies enhances the development of photovoltaic (PV) installation. In contrast, in [20], a model with PyPSA was developed to represent the historical curtailment in Germany. Their simulations contain spatial and temporal considerations. Results reveal that curtailment at high network resolution is considerably miscalculated. Nevertheless, as the network is clustered to a smaller number of nodes, an accurate model was obtained, reducing computation times and capturing the most important effects of network constraints.

These works have mainly focused on studying sector coupling without giving an overview of the interaction and flexibility between different technologies and the economic costs needed to achieve $CO_2$ emission reduction targets. In this sense, this paper proposes a restricted model of the German energy system for 2030, including the entire power sector and considering a part of the residential heat demand without including the transportation demand. The model was simulated under multiple defined $CO_2$ emission targets and included key technologies that couple different sectors. It aimed at studying the nexus among these technologies over a basis of one year, their flexibility, behaviour and capital cost under well-defined $CO_2$ emission targets.

This paper is organised as follows: Section 2 gives an overview of the technologies that have been included in this model, and Section 3 presents the model and the objective function, as well as the costs and the main assumptions within the model for the different sectors. In Section 4, the relevant findings are presented and discussed, while in Section 5, the limitations and boundaries of the model are explained. Finally, in Section 6, the main findings of the study are presented.

## 2. Technologies

Different technologies were employed in the model to meet various demands and fulfil the $CO_2$ reduction targets. Apart from the conventional power plants, the technologies considered in this study are grouped in the categories renewable power plants, sector-coupling technologies and storage technologies.

### 2.1. Renewable Power Plants

Germany's gross electricity consumption from renewables represented 41.1% of the total consumption in 2021. The wind dominated with a share of 20% of the total gross electricity consumption, followed by solar photovoltaics with 8.8%, biogas with 5% and hydropower with 4.3% [21].

#### 2.1.1. Wind

In the year 2021, 113.8 billion kWh were generated from wind. This shows a 14% decline as compared to the 132.1 billion kWh generated in 2020. The total installed capacity of wind technologies was 63.9 GW by the end of 2021 [21].

#### 2.1.2. Photovoltaic

PV had a 5 GW capacity installed in 2021, allowing the total installed capacity to reach 58.7 GW of solar PV [21].

#### 2.1.3. Biomass

Biomass and its derivatives represented almost 9% of the total electricity consumed in 2021 with 50.4 billion kWh of generation. Germany is a world leader in biogas production,

being the second-largest biogas producer in the world after the People's Republic of China [22].

*2.2. sector-coupling technologies*

There were a number of sector-coupling technologies that could be investigated in this study. In particular, to link the electricity sector with the heat sector, CHP plants, electric heating rods or heat pumps can be used.

In this study, heat pumps were chosen as a possible flexibility option because, in contrast to natural gas-fired CHP plants, they can also ensure the heat supply in zero-emission scenarios. At the same time, they have a higher efficiency than electric heating rods.

In order to link the gas sector with the electricity sector, electrolysers were examined in this study, as the use of hydrogen as a decarbonisation and storage medium is a key issue in the current debate.

### 2.2.1. Electrolysers

Hydrogen is a promising clean and sustainable energy carrier, because its use generates only water as a by-product and has no direct $CO_2$ emissions attached to it [23]. The federal government of Germany considers only green hydrogen—hydrogen produced from renewable energy—to be sustainable in the long run [24]. Green hydrogen can be produced by water electrolysis, which is the process of splitting water into hydrogen ($H_2$) and oxygen ($O_2$) using electricity. The unit in which the process takes place is referred to as an electrolyser [25]. Hydrogen plays a crucial role in coupling different sectors (power to gas), and it can additionally be an effective energy storage medium. With hydrogen, it is possible to store energy on medium- to long-term timescales, thus reducing energy curtailment and balancing the energy supply and demand [24].

### 2.2.2. Heat Pumps

Heat pumps (HPs) are another technology that are able to contribute to reducing GHG emissions as they can gradually replace the conventional fossil fuel boilers that are used in the heating sector today [26]. HPs use electricity and ambient heat to generate heat at a desired temperature; hence, they are the most important technology for coupling the electricity and heat sector. Moreover, HPs can also be used for the integration of renewable energy and a reduction in energy curtailment [13].

*2.3. Storage*

### 2.3.1. The Role of Storage Systems in the Energy Transition

Renewable energy systems are a vital component in achieving the desired goals of Germany in mitigating greenhouse gases and fulfilling the German demand for energy. Accordingly, the renewable energy capacities are expected to increase in the next decades. However, the integration of renewable energies in the power grid is connected to challenges in terms of stability and reliability as the renewable sources are intermittent by nature [27,28]. Even coastal winds vary during different weather conditions, and the solar output from solar systems fluctuates quite fast due to passing cloud patches in the sky [28]. The challenge of intermittent renewable energies can be addressed by integrating energy storage systems (ESSs) in the power grid [27]. EESs are essential to regulating fluctuating wind and solar energies [28] as they can add flexibility to the power system [27].

The increase in electricity curtailment from renewable energy sources creates a challenge for further deployment of these technologies [29]. Energy storage technologies could also be introduced to mitigate the limits of the transmission grid and reduce congestion and minimum generation constraints [29]. Thus, EESs are an integral part of a renewable energy system [28]. Germany is the biggest power market in Europe, and developing storage capacities alongside the renewable energy power generation could help meet the challenges associated with electricity generation from renewable energies and could have an impact beyond Germany's borders [30].

A wide variety of storage technologies are available to fulfil this task. These range from very efficient short-term energy storage, such as supercapacitors, to long-term energy storage using pumped hydro-storage power plants. As this study was conducted on an hourly resolution, the battery storage and already existing pumped hydro storage power plants were chosen as electrical storage technologies, as these two technologies best meet the flexibility requirements in an hourly resolution. To create a flexibility option for the other sectors, heat storage and hydrogen storage were also considered.

### 2.3.2. Hydrogen Storage

In the midst of the trend of decarbonisation, green hydrogen is seen as a clean energy carrier for cars as well as in marine traffic and could also serve as a very promising long-term energy storage solution for intermittent renewable energies [30]. In addition, there is an extensive usage of hydrogen in the industrial sector. Current and possible future use cases of hydrogen show the importance and the increased need for hydrogen storage with large capacities in the long-term. After the electrolysis process, the hydrogen is stored in low- or high-pressure vessels or in geological (underground) storage [30,31]. The geological underground caverns are the most suitable type for large capacity and long-term energy storing due to their large capacity potential and relatively high storage energy density [30,31]. The working theory behind underground storage is simple. It stores hydrogen under high pressure, allowing for it to be withdrawn when needed.

Such storage facilities can be integrated easily in rural or urban planning procedures; they provide secure and safe storage against potential threats such as fire, accidents or intentional damage (e.g., terrorist attack), which is an incredibly important feature for the energy security of supplies and unlike surface storage, which needs to occupy large surface areas of land. Economically, geological storages (e.g., salt caverns) are the best option when compared to other present alternatives [30].

### 2.3.3. Heat Storage

Combining renewable energies and storage systems for heat will play an essential role in transforming the heat energy sector [32]. In Germany, the primary uses of the residential heating sector for space heating and hot water production rely heavily on burning fossil fuels [22]. The integration of thermal energy storage (TES) will allow thermal energy collection from different sources independent of the demand and can be built at cost-effective prices [32].

There are various types of TES solutions, and they can be differentiated according to the characteristics of the stored material: sensitive, latent or thermochemical [32,33]. Likewise, the development of these technologies and their costs are at a different stage. TES based on sensitive materials already has a commercial use, with water being the most common seasonal storage material. By 2030, the cost is expected to decrease even further, its efficiency to increase and the installed capacity to grow significantly [34].

### 2.3.4. Battery Storage

The usage of battery energy storage systems is expected to have a high growth rate in combination with grid integration projects [35]. Batteries are one of the important technologies capable of providing the needed electric flexibility to overcome intermittent renewable power generation. The advantages of such systems include fast response times, a high efficiency, a low self-discharge rate and the feasibility of scaling due to the modular structure [35]. Today, the currently installed large-scale batteries (LSBs) in Germany are dominated by four types, namely, Li-ion, lead–acid, sodium–sulphur and redox flow [36].

### 2.3.5. Pumped Hydro Storage

Pumped hydro storage (PHS) is a mechanical storage system and the most widely used electricity storage system in Germany [37]. It is the most developed commercial storage technology and makes up about 94% of the world's energy storage capacity [27]. PHS

has a relatively high efficiency—it reaches efficiencies of up to 80%—and a long discharge duration, making it suitable for big energy applications [27,38]. PHS is typically established in mountain areas, creating controversies due to its substantial socio-economic-ecological impacts [38]. At the moment, there are only few suitable locations for the installation of additional PHS systems in Germany [38].

## 3. Methods

This section describes the energy system of Germany and the method used to conduct this study.

### 3.1. Model

The energy system was modelled to reveal the implications of the deployment of sector-coupling technologies in Germany. The modelled network, shown in Figure 2, was created using the tool python for power system analysis (PyPSA), for which all the input data were obtained using a literature review, as mentioned in this section, and are provided in Appendix A. The tool was accessed with the help of python, and the implementation of the model is available on GitHub at this link: https://github.com/dna-d8/Cire_project. The case study was carried out for the year 2030, and it encompassed the electricity sector, a part of the building sector and the hydrogen sector. In all sectors, the potential technology capacity was allowed to expand in order to meet the energy demand at each time step (one hour). In addition, the capacity optimisation was affected by marginal costs, capital costs, efficiencies and the global emission cap.

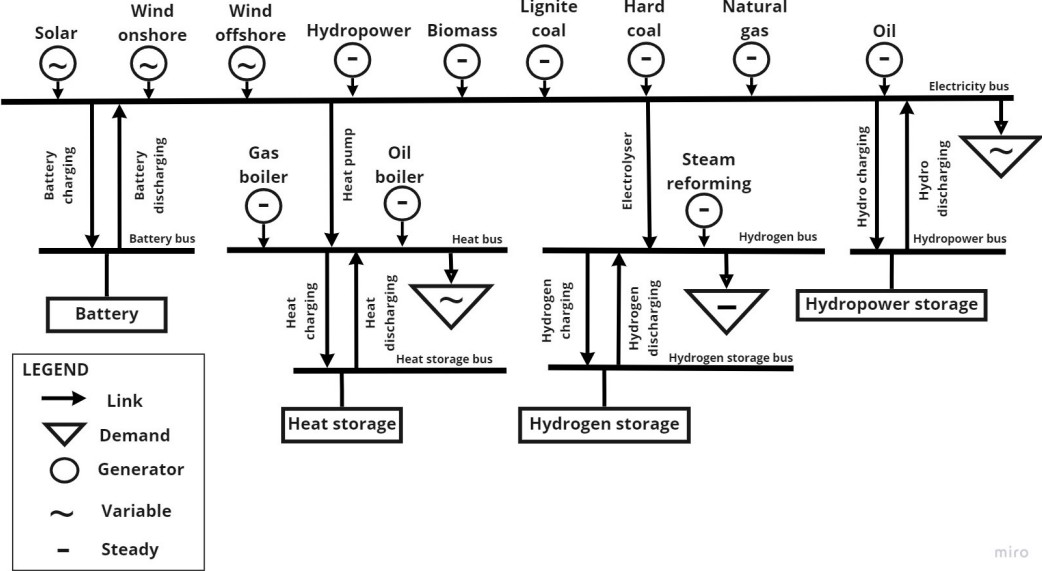

**Figure 2.** Germany's energy network was considered in the study with heat pumps, electrolysers, hydrogen storage, heat storage, battery storage and the electricity sector in 2030.

The system involved generation from renewable and fossil fuel energy sources to fulfil the energy demand of 2030. The technologies that are expected to play a major role in coupling the electricity and other sectors are power-to-heat (P2H) and power-to-hydrogen (P2H2) technologies such as heat pumps and electrolysers.

The following subsections describe the different sectors of the model in detail.

### 3.1.1. Electricity Sector

The electricity sector covers the total electrical load and generation in Germany. To resolve the annual values, an hourly time series was needed for the simulation of the load and the generation of intermittent renewable energies. The year 2018 was chosen as a base year for these time series. The historical data were derived from the Open Power

System Data project [39]. The profile for solar, offshore wind and onshore wind was considered without making any changes. However, the load of electricity had to be scaled up to 699 TWh, which represents the electricity demand in 2030 [40]. In the model, nine different technologies were considered for electricity generation; they can be divided into renewable and non-renewable resources. The initially installed capacities are shown in Figure 3.

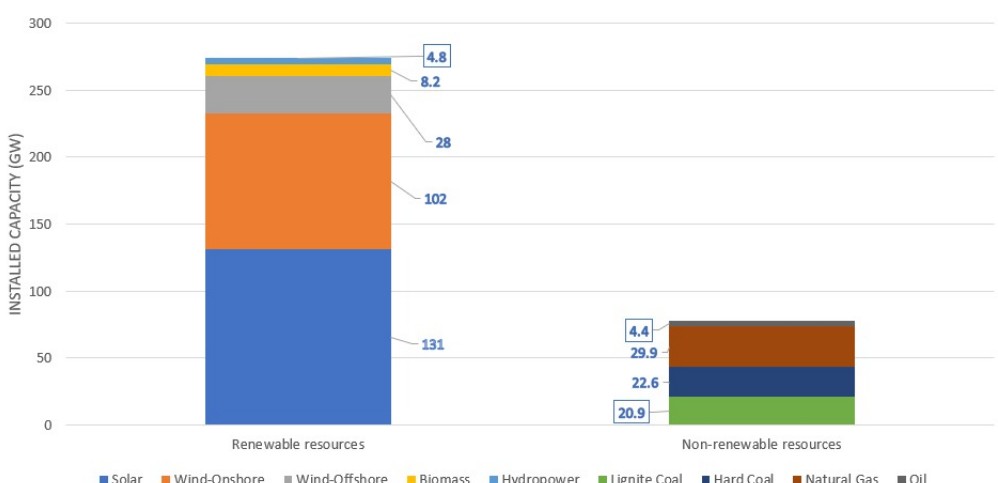

**Figure 3.** The expected installed capacity of electricity generation technologies in 2030 considered in the study, data obtained from [40].

Moreover, solar PV, onshore wind and offshore wind were allowed to expand in terms of installed capacity in order to reduce the $CO_2$ emissions; therefore, the capital costs of installing additional capacity of one $MW_{el}$ for the three technologies were calculated, as indicated in Section 3.3.2, based on the prices for 2030 as predicted by Sterchele et al. [41].

Another important factor for optimising the electricity system was the marginal costs [18]. They depend mostly on the fuel price and the generator efficiency, as discussed in Section 3.3.1; therefore, the marginal costs of the onshore wind , offshore wind, solar and hydropower technologies were assumed to be zero, while the marginal cost for biomass power plants was set to 120 EUR/MWhel based on one scenario for fuel prices and efficiency in 2030 [41]. The model was solved to minimise the total generation costs, including the capital and marginal costs for a desired $CO_2$ emission level. In the model, the $CO_2$ emission factors for all the renewable energy technologies were considered to be zero, while the $CO_2$ factor for the fossil fuels was obtained from [42].

### 3.1.2. Heating Sector

Around 60% of the heat used in Germany is used to heat buildings, while the remaining share is used in industrial processes. The primary use of the residential heating sector is for space heating and hot water production and relies heavily on burning fossil fuels [43]. It will be necessary to reduce the dependence on coal and oil by modernising old infrastructure, expanding heat generation based on renewable energies and coupling the heat and the power sectors through heat pumps. At the same time, gas will most likely remain important for the years to come.

The heating sector nevertheless will have to phase out oil and gas boiler capacities and increase the link with the power sector by increasing the installed capacity of heat pumps [44]. In 2015, the heat demand in Germany was about 730 TWh, with a high contribution from oil-fired boilers (25%) and gas-fired boilers (45%). In 2030, the coupling of the heating sector with electricity and the inclusion of efficient technologies could play an important role in reducing the dependency on fossil fuels [43]. This study focused on the impact of heat pumps on the building sector, with an hourly space and water heating

demand provided by [45]. Therefore, other sectors that use heat, such as district heat and industrial heat, were not considered.

### 3.1.3. Hydrogen Sector

In 2020, the total hydrogen production in Germany was around 57 TWh/year. Most hydrogen is produced as a by-product, e.g., in refineries or chloralkali processes. In refineries, most of the hydrogen is consumed in later processes. This model only considered hydrogen produced as the main product, for example, by steam reforming [46]. According to the national hydrogen strategy, 14 TWh of green hydrogen could be produced in Germany, and the total hydrogen demand will increase to a value of around 90–110 TWh/year. It is not expected that the whole demand will be fulfilled by domestically produced hydrogen [24]. Therefore, in the model, the hydrogen demand was assumed to be 32 TWh/year, which accounts for the sum of the total hydrogen produced by steam reforming in 2020 and the targeted green hydrogen production in Germany for 2030. The import of hydrogen was not in the scope of this study.

The hydrogen demand was considered to be constant over the whole year. Water electrolysis and natural gas steam reforming are the two technologies for producing hydrogen, both of which were considered in the model. The efficiency of the electrolysers (the higher heating value of the produced hydrogen divided by the consumed electricity [47]) considered in the model was 71%, which is the average efficiency of three water electrolysis technologies (alkaline electrolysis, AEL; high-temperature electrolysis, HTEL; and polymer replacement membrane electrolysis, PEMEL) [46]. The capital costs considered for electrolysers in this model were also the average of these three technologies. Further information on the cost calculation is presented in Section 3.3.2. The total installed steam reforming capacity was assumed to be 2.25 GWH$_2$ based on the 18 TWh of hydrogen produced in 2020 [46]. To calculate the installed capacity, the steam reformers were assumed to have 8000 full-load hours.

### 3.2. Optimisation Function and Constraints

To determine the optimal deployment and usage of the technologies in this model, PyPSA was used to simulate the model. PyPSA used the power flow equation to solve the model. It came with different options of optimisation tools, such as Gurobi and glpk. During solving this model, Gurobi was used as a solver to obtain the results. The optimisation function is defined as:

$$\min_{G_{n,r}, H^*_{n,s}, g_{n,r,t}} \left[ \sum_{n,r} c_{n,r} G_{n,r} + \sum_{n,r} c^*_{n,r} H^*_{n,r} + \sum_{n,r,t} o_{n,r} g_{n,r,t} \right] \tag{1}$$

$$\sum_r g_{n,r,t} = d_{n,t} \tag{2}$$

where $c_{n,r}$ is the fixed, annualised cost per capacity for a technology $r$ on bus $n$, and $G_{n,r}$ refers to the power capacity of generators and links. The $g_{n,r,t}$, dispatch of generators at the time $t$ is associated with $o_{n,r,t}$, the operation cost, and must be equal to energy demand $d_{n,t}$. $H^*_{n,r}$ is the energy capacity of a store, with the cost $c^*_{n,r}$ per storage capacity. These capacities were obtained while regarding the constraint to maintain a value between the minimum and maximum capacities, which is defined as follows.

$$G_{n,r_{min}} \leq G_{n,r} \leq G_{n,r_{max}} \tag{3}$$

$$H^*_{n,r_{min}} \leq H^*_{n,r} \leq H^*_{n,r_{max}} \tag{4}$$

The CO$_2$ boundary per investment period was implemented using a constraint, which is defined as:

$$\sum_{n,f,t} e_f \frac{g_{n,f,t}}{\eta_{n,f}} \leq CAP_{CO_2} \tag{5}$$

where $CAP_{CO_2}$ is the $CO_2$ emission cap with the emission factor $e_f$ in tonnes per MWh of a fuel $f$ and efficiency $\eta_{n,f}$ of the generators. A comprehensive overview of all equations used in the software can be found in [48]. Even though [48] presents the PyPSA-Eur model, the functions are identical to this study.

### 3.3. Costs

3.3.1. Marginal Cost

Marginal costs are the cost of generating one additional megawatt hour of electricity (MWh$_{el}$) [49]. In this model, marginal costs were based on the equation from [50], but only the fuel price and the generation efficiency were considered. All the other variables, including emission certificate prices, were neglected. As described in Section 3.4, the model was solved under several $CO_2$ emission boundaries; thus, the consideration of emission certificate prices would contradict the results.

$$MC = \frac{FP}{\eta_{th}} \tag{6}$$

where $MC$ are the marginal costs, $FP$ is the fuel price in EUR/ MWh$_{th}$, and $\eta_{th}$ is the generator efficiency (MWh$_{el}$/MWh$_{th}$).

3.3.2. Capital Cost

As the model represents one specific year, the investment costs must be taken into account by means of an annuity [18]; hence, different components with different lifetimes can be compared. According to [51], the equation to estimate the capital cost annuity is:

$$A = PV \frac{i}{1(1+i)^{-n}} \tag{7}$$

where $A$ is the annualised capital costs, $PV$ is the total investment costs, $i$ is the discount rate, which is assumed to be 2% according to [41], and $n$ is the lifetime.

### 3.4. $CO_2$ Scenarios

In this study, the $CO_2$ limits represent the emission cap on the modelled energy sectors in the year 2030. As described in Section 3.2, fossil-fuel-based technologies have emission carriers, and their usage is restricted through a total carbon cap of the whole system. The $CO_2$ limit starts from 250 Mt and is reduced in several scenarios in 10 Mt and 5 Mt decrements. The initial emission caps correspond to the emissions of the energy and building sectors in 2030 [52] and match moderately with the emissions of energy carriers foreseen for that year [5]. The $CO_2$ emission cap reached zero emissions in the last scenario. The result at each emission cap was obtained with the same input data and optimisation technique by introducing stricter carbon emission limits.

## 4. Results and Discussion

In this study, the system's adaptability to cope with tighter $CO_2$ emission limits was evaluated qualitatively and quantitatively. Figure 4 illustrates the different energy sources that meet the energy demand. The system evolved in response to various $CO_2$ scenarios, increasing installed renewable capacities and lowering the emission of fossil-fuel-based power plants. In this sense, the system can react flexibly by expanding storage and coupling the energy sectors. Flexibility, according to IRENA, is described as an electrical system's capacity to handle the fluctuation and uncertainty provided by renewable energies in various periods, from the very short to the very long term, while avoiding curtailment and reliably delivering the energy requested [53].

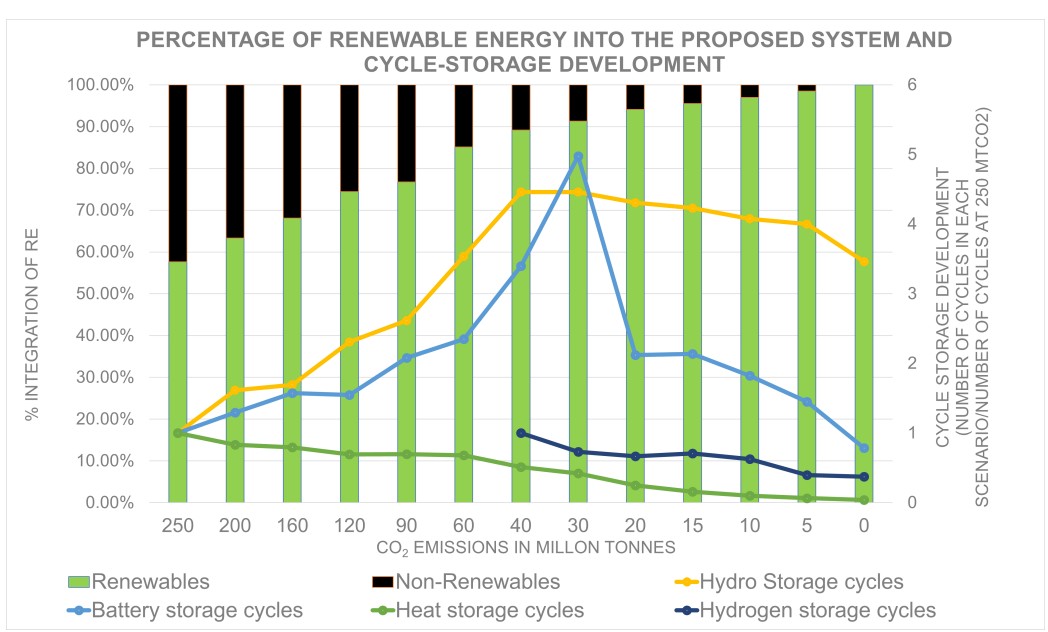

**Figure 4.** Involvement of renewable energy with restrictions on conventional sources and the development of storage cycles with different $CO_2$ emission levels.

In the German energy sector, a stricter $CO_2$ emission cap led to a higher expansion of renewable technologies, as shown in Figure 4. The capacities of these three generation technologies were almost three times higher when the system was made to be carbon-neutral.

The modelled hydrogen sector used steam reforming and electrolysers for hydrogen generation. Up to 40 million tons (Mt) of emissions, it can be noticed, were produced, even with the new steam reformers being implemented. In the same scenarios, the utilisation of existing electrolysers increased slowly to contribute to the tighter emission caps, as shown in Figure 5. Figure 6 shows that hydrogen storage became viable at the 40 Mt scenario, which also caused a sudden rise in electrolyser capacity and reduced the steam reforming capacity. With a significant RE penetration of about 90%, the usage rate of the electrolysers reached about 0.4. With more ambitious emission targets and higher renewable penetration rates, the usage rate of the electrolysers progressed higher, but the installed capacity started declining. Nevertheless, the total hydrogen production of the electrolysers increased with declining $CO_2$ emission levels.

Within the considered heat sector, heat pumps play a major role. Up to the 90 Mt scenario, their installed capacity was constant. The scenarios with higher $CO_2$ reduction ambitions showed a growing trend in heat pump installations. As Figure 7 shows, they doubled up to about 75 GW. However, during scenarios with high $CO_2$ savings and large RE penetration, the heat pump's usage rate was further reduced. At the same time, the installed capacities of heat pumps and storage were rising. This shows that the needed flexibility within the system is provided via intermittent operation of the heat pumps. The development of technologies in the individual sectors is shown in detail in Appendix B.

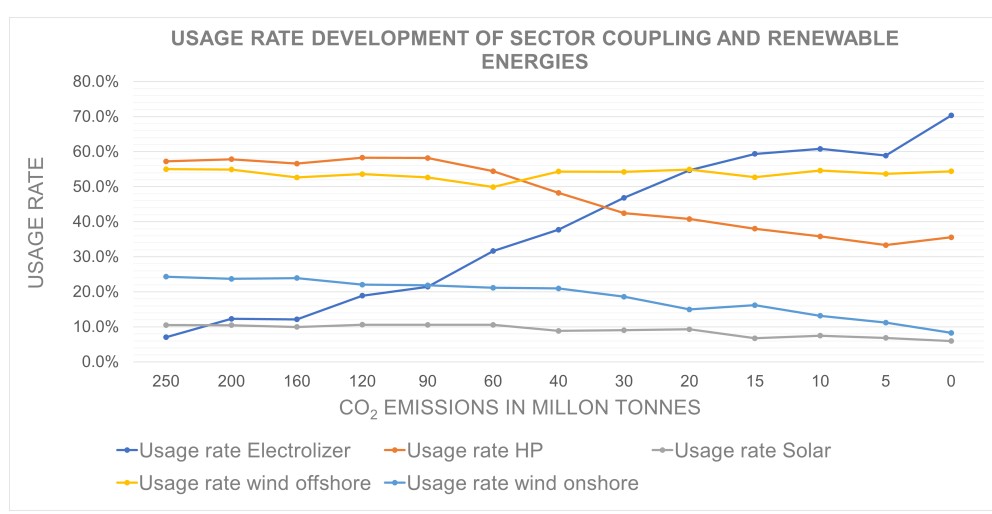

**Figure 5.** The usage rate of different technologies in the considered sector-coupled energy system with different $CO_2$ emission levels.

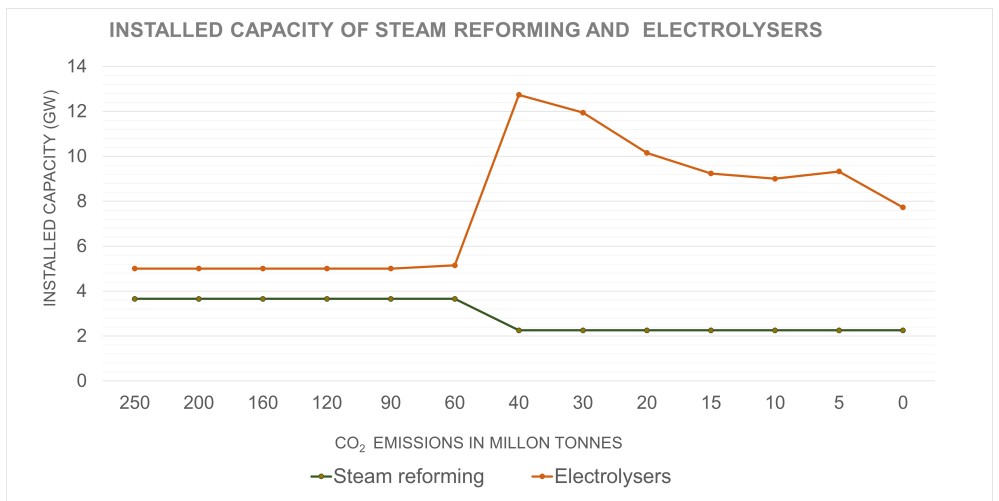

**Figure 6.** Development of the installed capacity of hydrogen production technologies considered in the model with different $CO_2$ emission levels.

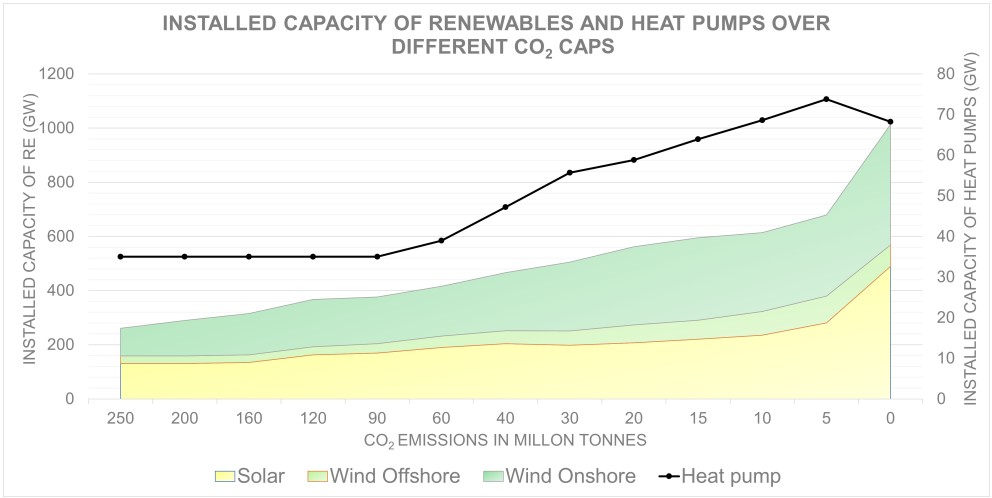

**Figure 7.** Development of the installed capacities of solar, onshore wind, offshore wind and heat pumps with different $CO_2$ emission levels.

With decreasing emissions and increasing RE and storage deployment, the overall system costs rose. Between the 0 Mt and the 250 Mt scenario, the costs were about 5.5 times higher. Figure 8 shows that the exponentially increasing costs did correspond with the rising amount of curtailed energy. Within the net-zero emissions scenario, the curtailed energy was even within the order of the electrical demand.

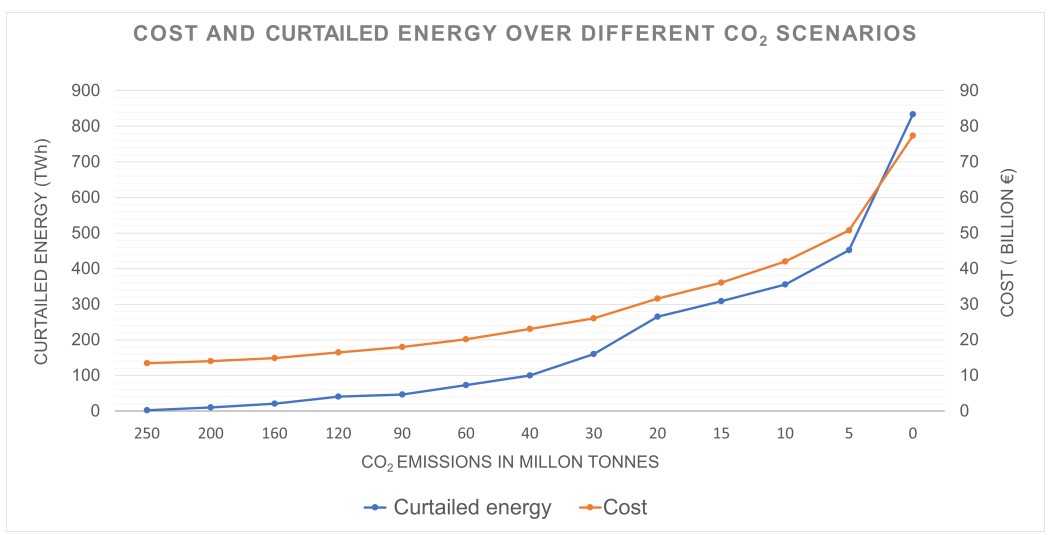

**Figure 8.** Electricity curtailment and the system's total cost considering capital and marginal costs with different $CO_2$ emission levels.

The usage rate of renewable technologies decreased with increasing curtailment. In Figure 7, the usage rate of offshore wind can be observed to be more or less stable. However, the onshore wind and solar usage rates decreased significantly in low-emission scenarios.

## 5. Limitations and Boundaries

The model proposed in this paper has various limitations. The system was insulated from the rest of the world, taking no electricity exchange into account. Also, the technologies were considered as a single unit. Section 3.1 considers and describes all power sources and demand by 2030. In this perspective, the electrical system was represented by a single bus containing all conventional and renewable energy sources that were used within the described energy system.

The model did not account for electric-vehicle-driven transportation demand, and only residential demand for water and space heating was considered when assessing the heating sector. Moreover, the model did not consider all hydrogen demand, and additionally, importing or exporting hydrogen was not considered. Energy storage, represented by batteries, hydro pumps, heat storage and hydrogen, increased the flexibility of the system. However, only those technologies that have reached market maturity or will have a significant impact by 2030 were considered in this study. The Section 2 provides a detailed review of the considered technologies.

Finally, the study did not aim to estimate the costs or the number of capacities required for the different technologies to reach a desired $CO_2$ level, but to show how the considered technologies behave and interact under different $CO_2$ emission caps in a sector-coupled energy system.

## 6. Conclusions

The study showed the behaviour of an energy system under different emission targets. It was noted that the usage rate of renewable technologies decreased, and the installed capacities continuously increased with higher limitations on emissions. The sector-coupling technologies behaved differently in this regard. Heat pumps showed similar behaviour, while electrolysers increased their usage rate as the RE penetration increased. The max-

imum installed capacity of electrolysers at the 40 Mt scenario showed that the system flexibility was not primarily maintained by the hydrogen sector but shifted towards the heating sector and electrical batteries. This can be explained by the high specific costs of the flexibility provided by electrolysers and is one core insight of this study. This contributes to the ongoing discussion about a comparison between flexibility options in the energy system.

Further research should explore the interplay between other sector-coupling technologies and flexibility opportunities. In addition, it might be helpful to investigate these technologies' integration into the European energy market and the development of the energy system over a period of at least two decades.

**Author Contributions:** Conceptualisation, A.N.; methodology, A.N.; software, D.P.; validation, A.S., D.P., J.C. and S.S.; formal analysis, D.P. and J.C.; investigation, A.S., J.C. and S.S.; data curation, D.P. and J.C.; writing—original draft preparation, A.S., D.P., J.C. and S.S.; writing—review and editing, A.N.; visualisation, J.C.; supervision, A.N. All authors have read and agreed to the published version of the manuscript.

**Funding:** This research received no external funding.

**Institutional Review Board Statement:** Not applicable.

**Informed Consent Statement:** Not applicable.

**Data Availability Statement:** Not applicable.

**Conflicts of Interest:** The authors declare no conflict of interest.

## Abbreviations

The following abbreviations are used in this manuscript:

| | |
|---|---|
| $CO_2$ | Carbon dioxide |
| EEG | German Renewable Energy Sources Act |
| ESSs | Energy storage systems |
| GHG | Greenhouse gas |
| $H_2$ | Hydrogen |
| HP | Heat pumps |
| IRENA | International Renewable Energy Agency |
| LSBs | Large-scale batteries |
| P2H | Power-to-heat |
| P2H2 | Power-to-hydrogen |
| PHS | Pumped hydro storage |
| PV | Photovoltaic |
| PyPSA | Python for power system analysis |
| RE | Renewable energy |
| TES | Thermal energy storage |

## Appendix A

The data used are presented in this section.

**Table A1.** Installed capacity, marginal cost, efficiency, capital cost and lifetime values of generators used in the model.

| Generators Type | Installed Capacity (MW) | Marginal Cost (EUR/MWh) | Efficiency | Capital Cost [1] (EUR/MW) | Lifetime |
|---|---|---|---|---|---|
| Solar | 131,000 [40] | 0 | 1 * | 718,000 [41] | 27 [41] |
| Offshore Wind | 28,000 [40] | 0 | 1 * | 2,937,000 [41] | 20 [41] |
| Onshore Wind | 102,000 [40] | 0 | 1 * | 1,366,000 [41] | 25 [41] |
| Biomass | 8200 [40] | 120 | 0.25 [54] | - | - |

**Table A1.** *Cont.*

| Generators Type | Installed Capacity (MW) | Marginal Cost (EUR/MWh) | Efficiency | Capital Cost [1] (EUR/MW) | Lifetime |
|---|---|---|---|---|---|
| Hydro-power | 4800 [40] | | 0.9 [55] | - | 80 [55] |
| Lignite Coal | 20,900 [40] | 5.1 | 0.39 [41] | - | - |
| Hard Coal | 22,600 [40] | 31 | 0.42 [41] | - | - |
| Natural gas | 29,900 [40] | 39 | 0.59 [41] | - | - |
| Oil | 4400 [40] | 137.8 | 0.37 [56] | - | - |
| Steam reforming | 2250 * | 27.7 | 0.83 | 300,000 [46] | 30 [46] |
| Heat boiler oil | 11,500 [57] | 54.3 | 0.94 [41] | 457,000 [58] | 20 [41] |
| Heat boiler gas | 58,000 [57] | 23.5 | 0.98 [41] | 387,000 [58] | 20 [41] |

1 Technologies without capital cost are assumed to be not extendable, * own assumptions

**Table A2.** Type of energy storage systems used in the model.

| Storage Type | Capacity (GWh) | Maximum Possible Capacity (GWh) | Capital Cost (EUR/MWh) | Life Time (Years) | Standing Losses/Day | Standing Losses/Hour |
|---|---|---|---|---|---|---|
| Battery | 26.1 [59] | - | 225,000 [60] | 15 [41] | 2% | 0.0833% |
| Hydro | 8000 [61] | - | 102,000 [30] | 80 [62] | 0.02% [62] | 0.0008% |
| Hydrogen | 0 * | 26,500 [63] | 6000 [30] | 30 [41] | 2% [63] | 0.083% |
| Heat | 54 [34] | - | 1945 [34] | 20 [34] | - | 0.0012% [34] |

* It was considered that in 2030 there will be no hydrogen storage capacity as a prerequisite, but that the needed capacity will be optimised based on the capital cost. Thus, the initial capacity was assumed to be zero for hydrogen charging and discharging.

**Table A3.** Price and emission factor for fuels used in the model.

| Fuel | Price (EUR/MWh$_{th}$) [42] | Emission Factor (Tons of $CO_2$ / MWh$_{th}$) [41] |
|---|---|---|
| Coal | 13 | 0.34 |
| Lignite | 2 | 0.4 |
| Natural gas | 23 | 0.2 |
| Oil | 51 | 0.28 |
| Biomass | 15 | 0 |

**Table A4.** Specification of links used in the model.

| Name | Efficiency | Capacity in (GW) | Capital Cost (EUR/MW) |
|---|---|---|---|
| Battery discharging | 0.872 [59] | Same as Storage capacity [59] | 23,750 [60] |
| Hydro charging | 0.792 [30] | 7.7 [64] | - * |
| Hydro discharging | 0.8 [64] | 7.7 [64] | - * |
| Hydrogen charging | 0.98 [31] | 0 | 128,000 [65] |
| Hydrogen discharging | 0.98 [31] | 0 | 41,121 [65] |
| Heat pump | 1.84 [43] | 35 [43] | 450,000 [41] |
| Heat storage charging | 0.989 [32] | 54 [22] | 2 [32] |
| Heat storage discharging | 0.989 [32] | 54 [22] | 2 [32] |
| Electrolyser | 0.71 [46] | 5 [24] | 1,137,000 [46] |

* Pumped hydro charging and discharging capital cost were not used in this study, as the pumped hydro technology was not considered to be extendable due to geographical and potential restrictions in Germany.

## Appendix B

The following figures show the supply and demand behaviour for the different energy sectors included in the model.

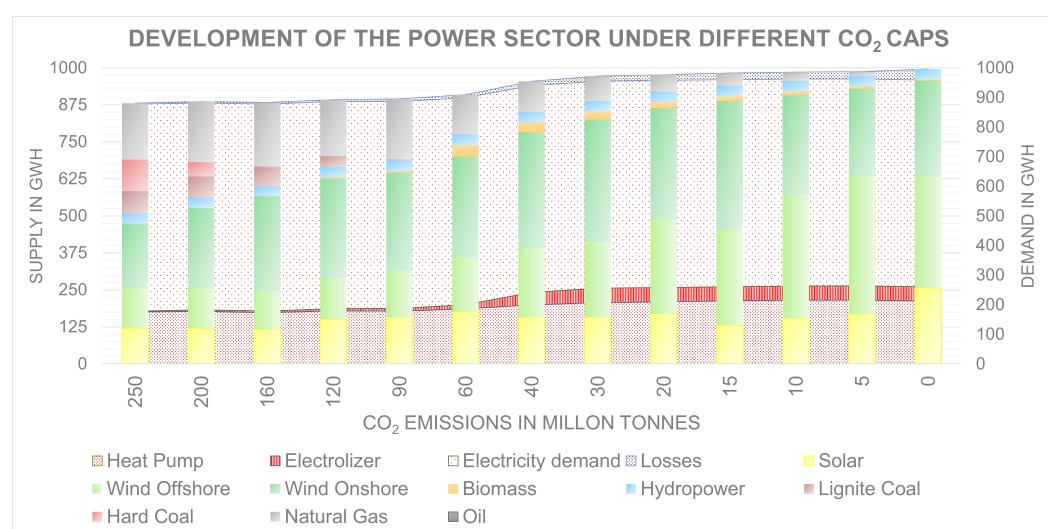

**Figure A1.** Development of the electricity sector under different $CO_2$ caps.

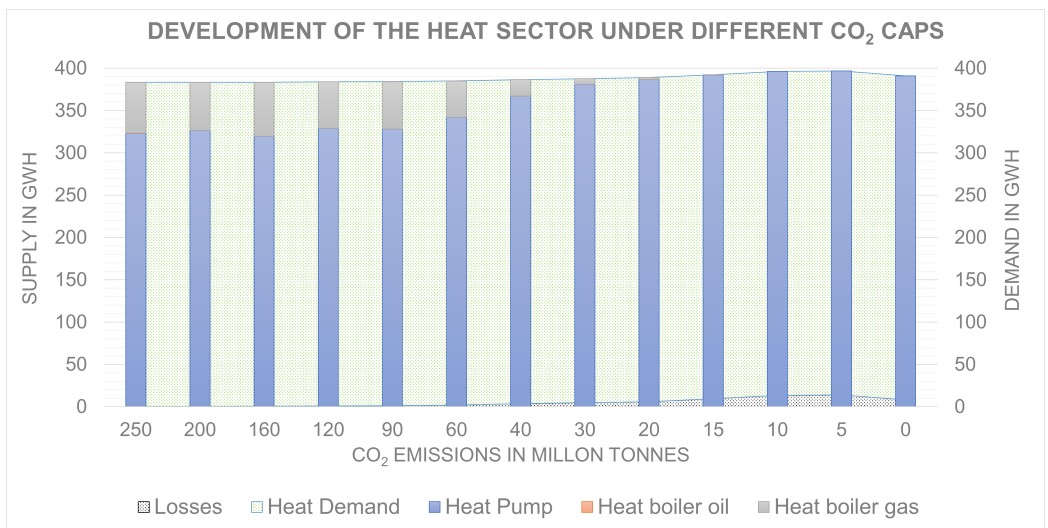

**Figure A2.** Development of the heat sector under different $CO_2$ caps.

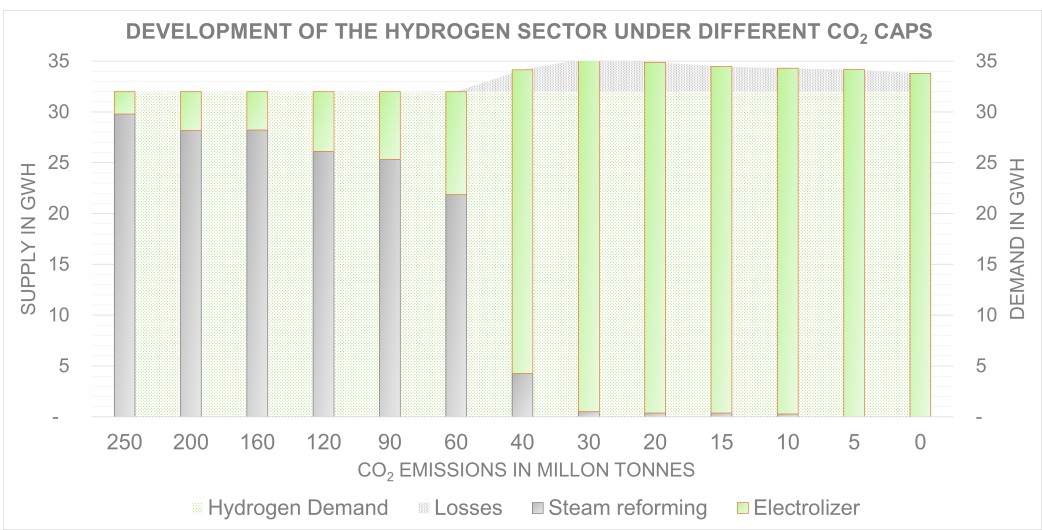

**Figure A3.** Development of the hydrogen sector under different $CO_2$ caps.

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
