# Peer review of "The Role of Renewable Energies, Storage and Sector-Coupling Technologies in the German Energy Sector under Different CO2 Emission Restrictions"

_sustainability, doi:10.3390/su141610379_

Round 1

Reviewer 1 Report

The article has a coherent and good structure. The process of data analysis is logical. This article needs minor modification.

It is better to emphasize the innovative aspects of this article in the abstract and summary.

Author Response

Dear Reviewer,

Thank you very much for your helpful review of our paper. We have taken your comments into account in the current version of the paper as follows:

  1. It is necessary to clearly state the innovative aspects of the research in the abstract and conclusion sections.
    The abstract and the conclusion have been improved and the unique contribution of this paper as well as the main insight has been added.
  2. It is necessary to give more details regarding the structure of the used software and the simulation process.
    The model description in section 3.1 has been improved with more details on the modelling as well as the discription of the optimisation function and the constrains in section 3.2 and the CO2 Scenarios in section 3.4
  3. How do we ensure the accuracy of simulation results? Is it possible to validate the results?
    Validating an energy model is difficult, especially if the model was designed for hypothetical scenario studies. However, the magnitudes of CO2 emissions (without limiting the emissions) are within the expected range of values for this particular system design. The same is true for the overall costs. At the same time, the input parameters taken from the literature have been compared with various sources and carefully checked.
  4. Only the heat pump and electrolyzer are considered in the coupling sector. Electrolyzers are used to produce hydrogen. A large part of hydrogen is consumed in the transportation sector. Ignoring the transport sector in this simulation can somehow affect the predicted behavior for electrolyzers. How can this contradiction be justified?
    The European Union has set specific targets for the inclusion of Hydrogen for final use. By 2030 it is expected that a large part of the final demand will take place in the industrial sector, since nowadays it is already used to produce goods in the chemical industry. While it is true that in the long term, energy sector players are pushing to shift final demand for fossil fuels, including transport, to green hydrogen, it is also worth noting that in this sector, part of the demand will be met by electric vehicles, as the infrastructure has been steadily developed in recent years and the overall efficiency of electric vehicles outweighs that of hydrogen-powered vehicles. 
    While the inclusion of different technologies and profile demands will directly impact the optimization function and ultimately the systems output, it is also noticeable that assumptions concerning model structure and input data are often based on the modellers’ subjective (Hofbauer, L et al, 2022). In this regard, the focus of the study is primarily in the evolution of the aforementioned coupling technologies under restrictive CO2 caps.
    https://www.fch.europa.eu/sites/default/file/Hydrogen%20Roadmap%20Europe_Report.pdf
    https://www.sciencedirect.com/science/article/pii/S1364032122002441
  5. To what extent does ignoring Germany's relations with the European Union and other countries of the world and ignoring exports and imports affect the results of the current study? Can the results based on these assumptions draw a correct vision of the future?
    Including other countries in the study and energy trade among them will allow it to have more flexibility and possibly reduce curtailed energy by allocating it better when necessary. Nevertheless, it was not the focus of the study to draw a correct vision of the future, but rather spot flexibility behaviors along selected technologies that could have a significant role in 2030.
  6. In your opinion, if we did the same simulation based on the data related to Germany in the previous 10 years for Germany in 2022, what results would we reach? How did the results correspond to today's reality?
    As this study focuses on the interaction of flexibility options in an economically optimised energy system under different CO2 constraints, the results would look quite similar to those in this paper. Only the difference in the input parameters, would lead to slightly different results. Especially the assumptions for costs of electrolysers, which are a pre-market technology, can change significantly within a decade. However, the main statements of this paper would probably remain the same.

Reviewer 2 Report

The paper analyzes Germany's renewable energy development and usage under different CO2 emission restrictions.

1. Minor editing comments:

fullfill -> fulfill 

electrolizer -> electrolyser (in the figure)

; instead of , in the keywords

Couple of keywords can be removed.

Subheadings could be used instead of bold titles.

Some references are in all-capitalized letters. Please modify.

A comprehensive proofreading is recommended.

2. Use of CHP as sector coupling between energy and heating sector has not been discussed:

https://www.sciencedirect.com/science/article/abs/pii/S1359431113009150

https://ieeexplore-ieee-org.proxy.lib.sfu.ca/document/8286154

3. Utilization of supercapacitors as energy storage has not been mentioned:

https://link-springer-com.proxy.lib.sfu.ca/article/10.1557/mrs.2012.222

4. "The model does not account for electric vehicle-driven transportation demand" 

"the electrical system is represented by a single bus"

"taking no electricity exchange into account"

These limitations significantly reduces the credibility of the work. Issues related to renewable energy penetration into the grid are ignored as a result of this simplification.

5. What are the major new contributions of this paper compared to the literature such as references 42 and 49 which seem to be more comprehensive?

Author Response

Dear Reviewer,

Thank you very much for your helpful review of our paper. We have taken your comments into account in the current version of the paper as follows:

  1. Minor editing comments:
    The editing comments have been corrected, spelling has been proofread again, the keywords are now updated, bold titles have been replaced by subheadings and all-capitilized references have been corrected.
  2. Use of CHP as sector coupling between energy and heating sector has not been discussed.
    Section 2.2 now discusses the sector coupling technologies with a special focus on CHP power plants.
  3. Utilization of supercapacitors as energy storage has not been mentioned
    Supercapacitors are now included in Section 2.3.1
  4. These limitations significantly reduces the credibility of the work. Issues related to renewable energy penetration into the grid are ignored as a result of this simplification.
    Since the focus of the study is to determine the flexibility behaviour of selected technologies under different CO2 emission constraints, these limitations only restrict the validity of the results to an acceptable extent. Nevertheless, taking an european energy market into account would surely reduce the calculated flexibility demand. Further research could be directed in this direction.
  5. What are the major new contributions of this paper compared to the literature such as references 42 and 49 which seem to be more comprehensive?
    This paper analyses the behaviour of flexibility options in a simply structured energy system under different CO2 emission constraints. The studies under 42 and 49 optimise a complex energy system towards a single CO2 emission target. 
    Our study can thus show the behaviour of flexibility options under different CO2 constraints and, in particular, how they behave non-linearly as the constraints are tightened. Such findings are not achievable with complex energy system models, as these have to be solved very frequently and would therefore be too computationally intensive.

Reviewer 3 Report

In this manuscript, the authors reported the functions of renewable energy sources, storage, and sector coupling technologies in the German energy sector under various CO2 emission restrictions and the findings of this study demonstrate how an energy system behaves when faced with various emission targets. The study is interesting. I recommend it accepted for publication after some revising. The main concerns are as follows:

1 All Figures need to be reconstructed. Hard to read captions of the figures. Tables format must match with journal style.
2 English expressions can be further improved.

Author Response

Dear Reviewer,

Thank you very much for your helpful review of our paper. We have taken your comments into account in the current version of the paper as follows:

  1. All Figures need to be reconstructed. Hard to read captions of the figures. Tables format must match with journal style.
    Caption contained in the figures were improved and they have now a better resolution which make them easily readable. Tables were updated in style and adapted to a wider format when necessary.
  2. English expressions can be further improved.
    English expression has been improved on various occasion, including the abstract, the introduction, the results, the limitations and the conclusion.

Round 2

Reviewer 2 Report

Authors addressed most of my concerns.